# Growth, Carcass and Meat Characteristics of Grass-Fed Lambs Weaned from Extensive Rangeland and Grazed on Permanent Pastures or Alfalfa

**DOI:** 10.3390/ani11010052

**Published:** 2020-12-30

**Authors:** Felipe Elizalde, Christian Hepp, Camila Reyes, Marilyn Tapia, Raúl Lira, Rodrigo Morales, Francisco Sales, Adrián Catrileo, Magdalena Silva

**Affiliations:** 1Centro Regional de Investigaciones INIA Tamel Aike, Instituto de Investigaciones Agropecuarias, Casilla, Coyhaique 296, Chile; chepp@inia.cl (C.H.); camila.reyes@inia.cl (C.R.); mtapia@inia.cl (M.T.); msilva@inia.cl (M.S.); 2Centro Regional de Investigaciones INIA Kampenaike, Instituto de Investigaciones Agropecuarias, Angamos, Punta Arenas 1056, Chile; rlira@inia.cl (R.L.); fsales@inia.cl (F.S.); 3Centro Regional de Investigaciones INIA Remehue, Instituto de Investigaciones Agropecuarias, Casilla, Osorno 24-0, Chile; rmorales@inia.cl; 4Medicina Veterinaria, Universidad Mayor, Av. Alemania, Temuco 281, Chile; adrian.catrileo@umayor.cl

**Keywords:** fatty acids, grass-fed meat, lambs, meat quality, pastoral systems

## Abstract

**Simple Summary:**

Meat consumers are nowadays more conscious about the effects of fatty acids on health, preferring meat with low saturated fat content and high levels of omega-3. Grass-fed ruminants have been shown to produce a more desirable fatty acid composition than those that are grain fed. Our study evaluates different lamb finishing systems. Overall, better sensory attribute scores were detected for lambs slaughtered at weaning. Meat from the three lamb production systems studied achieved health claimable levels of omega-3 fatty acids, especially in lambs slaughtered at weaning. There were no differences between the three systems of production for juiciness, however lamb finished on pasture presented a lower flavor score than animals finished on alfalfa or those slaughtered at weaning. The low level of cholesterol found in the three systems studied, along with low levels of total fat and saturated fat, classified these meats as extra lean. Opportunities exist for both, sheep producers and the industry, to take advantage of a more balanced fatty acid composition and health attributes of lamb meat produced in extensive farming systems. Developing finishing strategies on pastoral based systems is crucial to improve production characteristics, such as growth and carcass yield, without compromising the nutritive value of meat.

**Abstract:**

Western Patagonia lamb production systems are based on extensive rangeland. The harsh climate limits the adoption of more intensive systems. Therefore, producers must focus on developing differentiated products. The aim of this study was to evaluate growth patterns, carcasses and nutritive value of meat from 45 lambs. Lambs were divided into three groups: 15 lambs were slaughtered at weaning (W), with the remaining 30 weaned lambs being allocated to grazing either alfalfa (AG) or permanent pasture (PPG). AG lambs were significantly heavier and had higher condition scores than PPG lambs. Further, AG lambs showed higher carcass weight and larger tissue depth and commercial cuts. Moreover, W lambs had lower shear force and more tender meat than either AG or PPG lambs. The three systems showed a low shear force and acceptable sensory traits. Low levels of cholesterol, with low levels of saturated fat, classified these cuts of meat as extra lean. W lambs had higher omega-3 fatty acid content than AG or PPG lambs. Overall, results showed that meat from the three lamb production systems showed health claimable levels of omega-3 fatty acids and were low in fat and thereby, can be classified as lean meat.

## 1. Introduction

Lamb production in Western Patagonia (Chile) is based on grass, matching the growth curve of pastures. The main crop of lambs is harvested from extensive rangeland farms at weaning time, i.e., early summer, and delivers carcass weights in the order of 11–13 kg. Fatty acid composition of grass-fed meat has been seen as an opportunity for market advantages [1] and its nutritional value is an increasingly important factor influencing consumer preferences [2]. Lambs fed on steppe and highland pasture diets have been reported to have higher poly-unsaturated fatty acids content in their meat, especially in functional lipids influencing human health [3]. Several works highlight that ruminants fed on high grass-based diets provide milk and meat with a higher concentration of nutraceutical compounds [4]. Since forage is the most important component in ruminant diets in Patagonia, it is likely that nutritional strategies to influence fatty acid composition of beef and lamb meat are possible to apply [2,5].

Plant composition of species-diverse mountain grasslands can affect the sensorial and chemical attributes of milk and meat products, with implications for human health. Analyzing natural grasslands in the Italian alpine region, Enri et al. [6] found that the content of total fatty acids and that of the most abundant fatty acids (alfa-linolenic, linoleic and palmitic acids) were associated to nutrient-rich plant species, compared to dry grasslands. Scollan and Cooper [7] indicate that, when animals were grazing a legume-based pasture tended to produce heavy carcass weights, better conformation and a higher degree of fatness in comparison to animals grazing a grass-based pasture. According to Povolo et al. [8], variations in plant fatty acids and secondary plant metabolites, as well as grasslands with different botanical compositions, can confer specific intrinsic sensory and chemical attributes to dairy and meat products.

As forage availability of improved permanent grasslands occurs during summer in the lowlands of Western Patagonia, lambs weaned from tussock areas could graze these pastures in order to improve their performance. However, it is of importance to verify if this extended rearing period influences meat characteristics and quality, as well as fatty acid composition. The objective of this work was to compare lamb growth patterns, carcasses and nutritive value of meat from tussock weaned lambs grazing either improved permanent pasture or alfalfa pastures. Meat variables were also compared with traditional lambs slaughtered at weaning

## 2. Materials and Methods

### 2.1. Treatments and Animals

The work was conducted in accordance with requirements of the Chilean Law 20,380 on Animal Protection and with the approval of the INIA Bioethics Committee. A total of 45 male Corriedale lambs were raised to weaning with their dams on a typical sheep farm, representative of the main range farming area, located in the Aysén region of Western Patagonia. Ewes and lambs were grazed together on extensive rangeland, where the vegetation was dominated by *Festuca pallescens*, *Poa dussenni*, *Puccinellia* sp. and *Trifolium repens*. Weaning occurred on a single day when lambs averaged 27 kg live weight (LW) and 90 d of age. The lambs were then randomly allocated into three groups on the basis of their live weight (LW) and condition score [9]. The first group of 15 lambs was transported directly to a commercial abattoir, and their carcass characteristics at weaning (W) are presented in Table 1.

The remaining 30 lambs were transported to the Instituto de Investigaciones Agropecuarias, INIA Tamel Aike research station (45°45′ S, 72°02′ W, 480 m a.s.l.), located in Simpson Valley, Western Patagonia, Chile.

Lambs were assigned to two feeding groups: grazing on a legume-based pasture of alfalfa (*Medicago sativa*), (AG), *n* = 15, and grazing on improved permanent pasture, (PPG), *n* = 15. Improved permanent pasture was based on cocksfoot (*Dactylis glomerata*) and white clover (*Trifolium repens*). Animals were allocated to treatment groups based on LW. A rotational grazing was imposed which was controlled by an electric net fence. Lambs were shifted and weighed weekly during the experiment. Pasture was abundant (4700 and 3200 kg dry matter (DM) ha^−1^ on average for alfalfa and permanent pasture, respectively); net fence was moved when the residual was approximately 2500 and 1600 kg DM ha^−1^ for alfalfa and permanent pasture, respectively. There was a 7 d pre-experimental period to accustom animals to the two grazing systems and management conditions. No attempt was made to measure DM intake. At the beginning of the experiment, the lambs were treated with Fenbendazol 100mg (Panacur^®^ 10%), at 6 mg per kg, against liver, lung and intestinal parasites. Free access to watering was provided at all time. The experiment ranged from 21 January 2016 to 28 March 2016 (68 d) and lambs were slaughtered thereafter.

The botanical composition and chemical characteristics of improved permanent pasture and alfalfa consumed after weaning during the grazing period are shown in Table 2. For botanical composition, pasture samples from three 0.5 m^2^ frames were bulked and the proportions of botanical groups (grasses, legumes and herbs) as well as plant species within these groups were determined. Statistical evaluation of botanical analysis of pasture samples results was not performed as only an average (*n* = 3) pasture sample was available. For chemical characteristics, simulated grazed pasture samples were collected in polythene plastic bags from the allowance pasture strips. All the feed samples were transported under refrigeration to the laboratory and were dried for 48 h at 60 °C for chemical analyses. The chemical content of the feed samples was analyzed at the INIA Remehue Animal Nutrition and Environment Laboratory in Osorno, Chile.

The dry matter (DM), crude protein (CP), and ash were measured with the methods described by the AOAC [10]. The metabolizable energy (ME), in vitro digestibility (IVD) analysis and the neutral detergent fiber (NDF) were determined according to Sadzawka et al. [11].

### 2.2. Lamb Carcass Evaluation

In all cases, slaughter took place at the same local commercial abattoir, with pre slaughter live weights being recorded after 18 h of fasting. Lambs were stunned, and the carcasses were eviscerated and skinned according to normal commercial practices, weighed hot and held at 5 °C for 8 h. The hot carcass weight (HCW) included kidney-pelvic fat. The day after slaughter, cold carcasses were weighed (CCW) and graded for conformation using a four-point scale to represent very good, good, average or poor muscling, according to the Chilean standard for lamb carcasses [12]. Further slaughter data consisted of fat grade and kidney knob and channel fat grade (KKCF), using a three-point scale to represent deficient, normal or excessive fat covering. Carcass length from the last sacral vertebra to the first cervical (atlas) vertebra was also recorded.

Carcasses were split with a band saw between the 12th and 13th ribs and the fat depth over the eye muscle (*Longissimus dorsi*) was measured (mm) on both sides of the carcass with a sharpened steel ruler. Tissue depth (grade rule—GR) was also measured, as described by Kirton and Johnson [13]. Carcasses were then divided into commercial cuts according to the Chilean standard jointing procedure for lambs [14]. Commercial cuts were prepared and trimmed by experienced butchers and the trimmed cuts and trimmings were weighed. *Longissimus* muscles were vacuum packaged and transported to the meat laboratory of INIA-Remehue, Chile and kept frozen at −18 °C until analyses.

### 2.3. Color, pH and Texture

The Longissimus muscle samples were thawed for analysis for 48 h at 4 ± 2 °C, after which instrumental color was measured. The samples were then kept at room temperature for 30 min before measuring pH levels with a pH meter (HI 99163, Hanna Instruments, Madrid, Spain) with a penetration electrode (FC 232, Hanna Instruments). To analyze shear force, samples were cut into 3 cm samples and the external fat was removed. The samples were cooked in an oven at 170 °C until reaching a central temperature of 73 ± 1 °C (approximately 15 min). After being cooked, the samples were chilled at 4 ± 2 °C for 24 h. Some 6 to 10 1.3 cm circular specimens were extracted with a hollow punch to measure shear force with a texture analyzer (TA-XT2i, Stable Micro Systems, Godalming, UK) using the Warner–Bratzler method (WBSF) at a crosshead speed of 1 mm s^−1^, yielding the shear force (kgf).

### 2.4. Fatty Acid Composition

Samples of 10 g were used for the fatty acid analyses of meat, at the Animal Nutrition and Environment Laboratory of INIA Remehue in Osorno, Chile. Fat extraction method was performed according to Lumley and Colwell [15]. In the case of the meat, the trans methylation was done in accordance with Ichihara et al. [16]. After phase separation, the supernatant was collected and analyzed by gas chromatography. The FA profile was determined in a gas chromatograph (GC-2010 plus Shimadzu, Kyoto, Japan) equipped with a flame ionization detector. A capillary column SP-2560 (Sigma-Aldrich, Bellefonte, Pennsylvania, USA) of 100 m × 0.25 mm × 0.25 µm film was used. Helium was used as the carrier gas at 1.0 mL min^−1^ with an inlet pressure of 15 psi, using the split injection method (100:1). The injector temperature was fixed at 250 °C and the detector temperature at 260 °C. The injected sample volume was 1.0 µL and the oven temperature was programmed to increase from 140 (held for 5 min) to 240 °C (held for 15 min) at 4 °C min^−1^. Fatty acids were identified by comparing the retention times of the chromatograph peaks to those of the methyl esters from a mixture prepared with a 37-component FAME mix standard (Standard: 47885-U, Sigma-Aldrich Co, St. Louis, MI, USA), C18:1 t-11 methyl ester standard (Standard: 46905-U, Sigma-Aldrich) and c-9, t-11 octadecadienoic conjugated methyl acid (Standard: 10-1823-7, Larodan AB, Malmo, Sweden).

### 2.5. Sensory Analysis by Trained Panel

An 11-member trained panel participated in the sensory analysis. The panelists were selected from a group of 30 persons without previous experience in sensory evaluation. The training and testing sessions were conducted at the Sensory laboratory of INIA Remehue. The panelists completed 48 h of training sessions on the evaluation of a set of selected lamb meat attributes (flavor, tenderness and juiciness). The sensory laboratory was designed according to ISO standards with separate booths, and samples were evaluated in a sequence established to avoid the effect of sample order presentation, first-order or carry-over effects [17]. The panelists evaluated the three types of meat in duplicate. Cooked steaks were cut into dices of 20 mm × 20 mm × 25 mm (length/width/height) and kept warm. Subsequently, samples were placed in coded trays and served. The descriptors were quantified using a hybrid scale ranging from 0 (absence) to 10 (maximum intensity) [18].

### 2.6. Statistical Procedures

The data were subjected to ANOVA to examine the effect of grazing group (two treatments after weaning) on lamb performance, carcass characteristics and meat cut characteristics. For the chemical and physical characteristics and FA composition of meat, comparisons were made between the three production systems (including the “at weaning” treatment). Comparisons of discrete variables (KKCF grade, fat grade and conformation classification) were performed using Kruskal–Wallis non parametric test. These data were analyzed using the General ANOVA procedure in InfoStat Software (Facultad de Ciencias Agropecuarias, Universidad Nacional de Córdoba, Argentina), considering a completely randomized design, treatment differences were tested for significance at the 0.05 probability level based on the Fisher’s least significant difference (LSD) method. The ANOVA for sensory data (tenderness, juiciness and flavor) were performed with the General Linear Model (GLM) procedure of the SAS system (SAS Inst. Inc., Cary, NC, USA). Treatments were used as fixed effects and differences among effects were tested using Tukey’s test.

## 3. Results

The effects of post-weaning lamb fattening system on LW gain and condition score for the two feeding groups: grazing on a legume-based pasture of alfalfa (Medicago sativa), (AG) and grazing on improved permanent pasture, (PPG), are presented in Table 3.

Lamb LW gain was higher (*p* < 0.001) in AG lambs, compared to PPG lambs, therefore AG lambs averaged a 4.3 kg heavier body weight (*p* < 0.001) than PPG lambs. Condition score classification followed a similar pattern, with AG lambs presenting a significantly higher condition score (*p* < 0.001) than PPG lambs.

### 3.1. Carcass Evaluation

Lamb carcass characteristics at weaning, and the effects of two post-weaning lamb fattening systems on these traits are presented in Table 4.

Alfalfa grazed lambs reached a significantly higher HCW (*p* < 0.001) and CCW (*p* < 0.001) compared to PPG lambs. Furthermore, AG lambs had significantly (*p* < 0.05) larger tissue depth (GR) than PPG grazed lambs. The fattening system did not alter significantly dressing percentage, carcass length, KKCF, fat grade and conformation classification measurements (*p* > 0.05).

The different cuts of meat (Table 5) were all significantly affected (*p* < 0.05) by the type of pasture to a lesser or greater extent, except for rump bone and residues, as a result of the PPG lambs being smaller and having a lighter carcass than AG lambs.

The chemical and physical characteristics of lambs fattened on different pasture systems are presented in Table 6. A significantly lower (*p* < 0.05) dry matter content of the meat from PPG lambs was detected, compared with animals slaughtered at weaning (W) or AG lambs.

There were no significant effects (*p* > 0.05) of production systems on protein content, ether extract, pH and muscle color parameters. Lambs slaughtered at weaning had greater (*p* < 0.05) ash content than AG lambs. Further, lambs slaughtered at weaning presented lower (*p* < 0.05) shear force than either AG or PPG lambs.

### 3.2. Sensory Panel

The effects of system of production on three sensory traits are presented in Table 7. Lambs slaughtered at weaning presented a more (*p* < 0.05) tender meat than lambs finished on permanent pasture, with alfalfa grazed lambs being intermediate for tenderness. There were no differences (*p* > 0.05) between the three systems of production for juiciness, however lambs finished on permanent pasture presented a lower (*p* < 0.05) flavor score than lambs finished on alfalfa or slaughtered at weaning.

### 3.3. Fatty Acid Composition

Fatty acid content of Longissimus muscle (mg 100 g^−1^) is presented in Table 8. There were no significant differences (*p* > 0.05) between the three lamb production systems in terms of their saturated and monounsaturated fat content, ranging between 1295 and 1362 mg 100 g^−1^ for saturated fat content and between 946 and 1033 mg 100 g^−1^ for monounsaturated fat content. However, there was a significant effect in (*p* < 0.05) polyunsaturated fatty acid (PUFA) content in lambs slaughtered at weaning, in relation to animals finished on alfalfa or permanent pasture. It is important to note a higher (*p* < 0.05) linoleic content on animals slaughtered at weaning in comparison to animals finished on permanent pasture, with animals finished on alfalfa being intermediate for this trait.

Furthermore, lambs slaughtered at weaning had a significantly higher (*p* < 0.05) content of alpha-linolenic acid, eicosapentaenoic acid (EPA), docosahexaenoic acid (DHA) and overall omega 3 content, than lambs finished on alfalfa or permanent pasture.

A lower (*p* < 0.05) cholesterol level was detected when animals where finished on permanent pasture, in comparison with the other two groups of lambs.

## 4. Discussions

### 4.1. Lamb Growth Characteristics

In the present study, lambs finished on alfalfa were heavier than lambs finished on permanent pasture, in line with previous reports in the literature. For example, McCutcheon [19] demonstrates that lambs finished on ryegrass took longer to reach the target weight than lambs finished on alfalfa.

Lambs finished on alfalfa had a greater offer of nutrients (Table 2), with more protein than those on permanent pasture. As pointed out by Danso [20], protein requirements for growing lambs vary from 125 g/kg DM intake to approximately 175 g/kg DM intake for weaned lambs greater than 20 kg LW, therefore it is suggested that, in this experiment, when grazed on permanent pasture, the lower protein consumption would be reflected in a lower weight gain

It is important to note that during the summer season 2016, the rainfall dropped to only one third of the average rainfall—the greatest drought recorded during the last 50 years [21]. This drought had a great influence on the availability of fresh grass, especially permanent pastures; whereas alfalfa, although affected, showed a better tolerance. This lack of nutrients is likely to be the main cause of poor performance on PPG lambs.

### 4.2. Lamb Carcass Characteristics

The positive effects of using legume-based pastures on lamb growth rates, carcass weight and overall animal response have been extensively reported in the literature [22].

The present study showed that the CCW of lambs finished on alfalfa was 1.8 kg heavier compared to lambs finished on permanent pasture. In addition, the 68 d younger group of animals slaughtered at weaning showed a similar CCW than PPG lambs. The low LW gain in PPG animals and the drop in dressing percentage for the post weaned treatments [23] might explain the poor response recorded for this group of lambs.

As suggested by Bianchi [24], the optimum tissue depth (GR) for light lamb carcasses (10–14 kg) should vary between 5 and 7 mm. In the current study, lambs slaughtered at weaning and animals finished on alfalfa met this criterion, meaning a proper finishing, whereas PPG lambs did not, suggesting the need for protein supplementation in order to achieve a higher tissue depth.

### 4.3. Chemical and Physical Characteristics

In Norway, Adnoy et al. [25] found small but significant differences in physicochemical characteristics of meat from groups of lambs grazing unimproved mountain pastures compared to fertilized cultivated lowland pastures during a two-month period prior to slaughter. In the present study, only small differences were detected between the three groups of lambs for these traits. Dry matter content of animals slaughtered directly at weaning was lower than the other two groups of lambs, probably due to the lower age at slaughter, reflecting lower connective tissue strength. Regarding the measures in color, it is worth mentioning that in all three cases, the lightness (L*) value was above 34, and when the degree of redness (a*) value is equal to or exceeds 9.5, on average consumers will consider the lamb meat color acceptable [26].

Australian and New Zealand consumers accept lamb tenderness at about 2.7 kgf [27] but if the value is ≥ 6 kgf it is considered unacceptable [28]. Overall shear force recorded for the three groups were well below these values for this parameter. Although it is worth noting that the younger group of lambs, slaughtered at weaning, showed a significantly lower shear force compared to the other two groups.

### 4.4. Sensory Characteristics

In the present study, the meat of lambs slaughtered at weaning, straight from rangeland, was more tender than when lambs were finished on permanent pasture, with alfalfa grazed lambs being intermediate, suggesting that tenderness might decline with age, as pointed out by Garza [29]. However, higher GR values for W and AG lambs suggested higher carcass fatness [13], and since flavor profile of meat can be expressed differently based on the amount and proportion of volatile compounds found in lipids, the differences of flavor detected might partly be attributed to this. Furthermore, it has been reported that lamb meat from natural pastures was tenderer than meat from lambs finished on cultivated pastures [25,30], as was the case in this study, with the lower shear force detected for the group of animals slaughtered at weaning.

On the other hand, no differences for juiciness were detected between the three systems of lamb production. Previous work has reported a similar trend for this trait. For example, Fisher et al. [31], using Suffolk lambs, found no differences in terms of juiciness when comparing two finishing systems, although there were significant differences when the comparison was made on the basis of different breeds.

In opposition to previous work done in New Zealand [32], the current study showed that lambs finished on permanent pasture, dominated by cocksfoot, resulted in a lower flavor score, in comparison to lambs slaughtered at weaning or finished on alfalfa. This might be partly attributed to the lower final weight of the former group of lambs, in relation to the group finished on alfalfa.

Overall, high sensory attributes were detected by taste panel at weaning in comparison to those lambs grazed post weaning on permanent pasture. A favorable response on tenderness and flavor was found on these traits, in relation to animals finished on permanent pasture.

### 4.5. Fat Content and Fatty Acid Composition

In the current experiment, although no differences between the three production systems in terms of total saturated or monounsaturated fat content were detected, a higher concentration of 18:3 (alpha-linolenic acid), EPA, DHA and overall long chain n-3 PUFA content was measured on lambs fed on natural rangeland and slaughtered at weaning, in relation to animals finished on alfalfa or permanent pasture. It has been pointed out, that the type of diet has a significant effect on PUFA content of the resulting meat [33]. Previous studies [25,30] found that, when comparing meat from lambs finished on different pastures, only relatively small differences in the fatty composition are found.

Functionally, the two most important omega-3 fatty acids are EPA (20:5n-3) and DHA (22:6n-3), however, docosapetaenoic acid (DPA; 22:5n-3) has also been shown to have an important role [34]. According to Australian and New Zealand standards [35], to claim meat as a source of omega-3, it needs to have 30 mg long chain omega-3 FA per serving of meat in the form of EPA or DHA. Previous studies [22] consider 135 g meat to be a serving, therefore lamb with EPA plus DHA > 23 mg 100 g^−1^ meat will meet this standard.

When considering the standard portion of 135 g fresh meat, EPA plus DHA content recorded in this study for the three systems were 42.1; 32.1, and 31.6 mg per serving for lambs slaughtered at weaning, finished on alfalfa and finished on permanent pasture, respectively. This indicates that meat from the three lamb production systems studied in our region, achieved levels of omega-3 fatty acids that can be claimed as healthy, especially in lambs slaughtered at weaning.

The low levels of cholesterol found in the three systems studied, along with low levels of total fat and saturated fat, classified these cuts of meat as “extra lean”, according to the Chilean legislation [36].

## 5. Conclusions

Given that the demand for healthy foods in the world is increasing, and that pastoral based feeding systems are viewed by consumers as more natural and friendly, the outcomes of this study are important for the marketing of grass-fed lamb meat. The three pastoral systems studied showed a low shear force and acceptable sensory traits. Lamb meats produced under the three pastoral systems studied were low in fat, and they can be also classified as lean meat according to Chilean standards. Overall, the meat produced on the three pastoral based feeding systems is considered appropriate for human health, with an especially higher content of EPA and DHA. It is however important to note that these results are for lamb loins, and it would be beneficial to assess whether these characteristics apply to the rest of the commercial cuts. On the other hand, much care should be taken when taking the decision to finish the lambs on grass after weaning, on pasture, if food available is in short supply thereafter.

## Figures and Tables

**Table 1 animals-11-00052-t001:** Lamb liveweight and carcass characteristics at weaning.

	W
Live weight (kg)	28.1
Hot carcass (kg)	12.7
Cold carcass (kg)	12.2
Dressing percentage (%)	45.1
Carcass length (cm)	59.3
KKCF grade ^1^	1.67
Conformation class ^2^	2.33
GR (mm)	7.10

^1^ KKCF = kidney knob and channel fat grade (deficient = 1 and excessive = 3); ^2^ Chilean conformation classification (very good = 1 and poor = 4); GR = grade rule.

**Table 2 animals-11-00052-t002:** Botanical composition and chemical characteristics of the pasture consumed in the experiment at the start and end of the grazing period.

Variable	Type of Pasture			
	Permanent Pasture		Alfalfa	
	Initial	Final	Initial	Final
Botanical composition (%)				
*Dactylis glomerata*	53.8 ± 11.0	43.6 ± 5.9		
*Poa pratensis*	9.1 ± 1.9	4.4 ± 0.8		
*Trifolium repens*	5.1 ± 1.7			
*Holcus lanatus*	0.6 ± 0.6	1.1 ± 0.3		
*Medicago sativa*			66.1 ± 9.1	11.0 ± 1.6
*Taraxacum officinale*			3.5 ± 1.1	0.0 ± 0.0
Other grasses			11.6 ± 5.4	20.7 ± 4.0
Weeds	5.8 ± 1.3			
Dead material	25.6 ± 10.5	51.0 ± 6.4	18.9 ± 3.6	68.3 ± 3.2
Bromatological composition				
DM (%)	42.5 ± 0.03	56.6 ± 0.45	26.3 ± 0.80	67.4 ± 2.30
CP (%)	9.4 ± 0.23	6.3 ± 0.41	18.4 ± 1.42	10.8 ± 0.7
IVD (%)	65.2 ± 1.43	64.2 ± 0.12	68.5 ± 1.82	58.1 ± 0.8
ME (Mcal kg^−1^)	2.2 ± 0.04	2.2 ± 0.00	2.2 ± 0.05	2.0 ± 0.01
NDF (%)	52.6 ± 1.79	60.9 ± 0.87	39.2 ± 1.87	56.1 ± 0.7
WSC (%)	11.9 ± 0.18	11.8 ± 0.45	7.1 ± 0.42	9.1 ± 1.3

DM = dry matter; CP = crude protein; IVD = in vitro digestibility; ME = metabolizable energy; NDF = neutral detergent fiber; WSC = water soluble carbohydrates.

**Table 3 animals-11-00052-t003:** Effects of pasture type on lamb performance.

Variable	AG	PPG	SEM	Significance of *p* Value
Initial live weight (kg)	27.1	26.9	0.199	Ns
Live weight gain (kg d^−1^)	0.17 ^a^	0.10 ^b^	0.010	***
Final weight (kg)	38.4 ^a^	34.1 ^b^	0.595	***
Body condition score	4.1 ^a^	3.2 ^b^	0.120	***

SEM = Standard Error of the Mean; Ns = not significant; ^a,b^ values within rows without common superscript letters differ (at least *p* < 0.05). Significantly different: *** *p* < 0.001; Ns: *p* > 0.05.

**Table 4 animals-11-00052-t004:** Effects of pasture type on lamb carcass characteristics.

	AG	PPG	SEM	Significance of *p* Value
HCW (kg)	14.2 ^a^	12.4 ^b^	0.23	***
CCW (kg)	13.9 ^a^	12.1 ^b^	0.22	***
Dressing percentage (%)	37.1	36.2	0.53	Ns
Carcass length (cm)	63.5	62.1	0.38	Ns
KKCF grade ^1^	1.93	1.53	0.18	Ns
Fat grade ^2^	2.40	2.27	0.18	Ns
Conformation class ^3^	2.60	2.87	0.15	Ns
GR (mm)	5.00 ^a^	3.50 ^b^	0.40	*

SEM = Standard Error of the Mean; Ns = not significant; KKCF = kidney knob and channel fat grade; GR = grade rule; ^a,b^ values within rows without common superscript letters differ (at least *p* < 0.05); ^1^ KKCF (deficient = 1 and excessive = 3); ^2^ fat grade (deficient = 1 and excessive = 3); ^3^ Chilean conformation classification (very good = 1 and poor = 4); significantly different: * *p* < 0.05; *** *p* < 0.001; Ns: *p* > 0.05.

**Table 5 animals-11-00052-t005:** Effects of pasture type on lamb meat cut characteristics.

Variable (kg)	AG	PPG	SEM	Significance of *p* Value
Backstrap fillet	0.35 ^a^	0.29 ^b^	0.01	***
Leg	3.15 ^a^	2.66 ^b^	0.07	***
Shoulder	2.33 ^a^	2.00 ^b^	0.07	***
Forequarter rack	1.00 ^a^	0.80 ^b^	0.03	***
Lamb rack	1.25 ^a^	1.05 ^b^	0.03	***
Tenderloin	0.08 ^a^	0.07 ^b^	0.00	*
Neck	0.85 ^a^	0.75 ^b^	0.02	**
Flap	1.96 ^a^	1.67 ^b^	0.04	***
Hind shank	0.67 ^a^	0.61 ^b^	0.01	**
Rump bone	1.09	1.01	0.05	*p* = 0.074
Residues	1.18	1.17	0.06	Ns

SEM = Standard Error of the Mean; Ns = not significant; ^a,b^ values within rows without common superscript letters differ (at least *p* < 0.05). Significantly different: * *p* < 0.05; ** *p* < 0.01; *** *p* < 0.001; Ns: *p* > 0.05.

**Table 6 animals-11-00052-t006:** Chemical and physical characteristics of lambs fattening on different production systems.

	W	AG	PPG	SEM	Significance of *p* Value
Dry matter (%)	23.9 ^a^	23.5 ^a^	22.3 ^b^	0.34	**
Protein (%)	81.6	82.9	83.7	0.64	*p* = 0.084
Ash (%)	4.5 ^a^	3.7 ^c^	3.9 ^b^	0.06	***
Ether extract (%)	10.5	13.6	11.9	1.02	Ns
pH	5.8	5.7	5.7	0.04	Ns
*L**	38.0	38.8	38.6	0.46	Ns
*a**	17.9	17.2	17.3	0.39	Ns
*b**	8.7	17.2	9.0	0.32	Ns
Shear force (kgf)	1.6 ^b^	2.1 ^a^	2.2 ^a^	0.11	***

SEM = Standard Error of the Mean; Ns = not significant; ^a,b,c^ values within rows without common superscript letters differ (at least *p* < 0.05). Significantly different: ** *p* < 0.01; *** *p* < 0.001; Ns: *p* > 0.05.

**Table 7 animals-11-00052-t007:** Sensory attributes of lambs fattening on different production systems.

	W	AG	PPG	SEM	Significance of *p* Value
Tenderness	8.42 ^a^	7.68 ^a,b^	7.43 ^b^	0.27	*
Juiciness	5.11	4.64	4.77	0.23	Ns
Flavor	6.26 ^a^	6.21 ^a^	5.68 ^b^	0.17	*

SEM = Standard Error of the Mean; Ns = not significant; ^a,b^ Values within rows without common superscript letters differ (at least *p* < 0.05). Traits were quantified using a hybrid scale ranging from 0 (absence) to 10 (maximum intensity). Significantly different: * *p* < 0.05; Ns: *p* > 0.05.

**Table 8 animals-11-00052-t008:** Fatty acid content (mg 100 g^−1^) of longissimus muscle in three production systems.

	W	AG	PPG	SEM	Significance of *p* Value
Palmitic 16:0	617.0	647.7	612.7	46.60	Ns
Stearic 18:0	403.5	456.2	440.3	28.69	Ns
Trans-vaccenic 11t-18:1	57.7	60.7	53.7	5.36	Ns
Oleic 9c-18:1n-6	705.9	796.6	751.6	54.50	Ns
Linoleic 18:2n-6	131.3 ^a^	117.9 ^a,b^	112.1 ^b^	5.35	*
α linolenic 18:3n-6	57.1 ^a^	45.5 ^b^	42.4 ^b^	3.07	**
CLA Rumenic 9c,11t-18:2	26.0	24.6	21.9	2.48	Ns
EPA 20:5n-3	24.6 ^a^	18.5 ^b^	18.3 ^b^	1.00	***
DHA 22:6n-3	6.6 ^a^	5.3 ^b^	5.1 ^b^	0.32	**
DPA 22:5n-3	21.2	19.4	19.1	0.87	Ns
Total saturated	1303.1	1361.5	1294.8	93.99	Ns
Total monounsaturated	945.5	1033.3	970.0	69.47	Ns
Total polyunsaturated	298.7 ^a^	261.3 ^b^	252.4 ^b^	10.72	**
n-6	178.0	163.1	157.6	6.58	*p* = 0.087
n-3	112.1 ^a^	90.6 ^b^	87.1 ^b^	4.56	***
Cholesterol	47.3 ^a,b^	50.0 ^a^	43.0 ^b^	1.75	*
Total CLA	32.0	30.5	27.5	2.95	NS

SEM = Standard Error of the Mean; Ns = not significant; CLA = conjugated linoleic acid; EPA = eicosapentaenoic acid; DHA = docosahexaenoic acid; DPA = docosapentaenoic acid. ^a,b^ Values within rows without common superscript letters differ (at least *p* < 0.05). Significantly different: * *p* < 0.05; ** *p* < 0.01; *** *p* < 0.001; Ns: *p* > 0.05.

## Data Availability

The data can be available from the corresponding author upon reasonable request.

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
