# Peer review of "Growth, Carcass and Meat Characteristics of Grass-Fed Lambs Weaned from Extensive Rangeland and Grazed on Permanent Pastures or Alfalfa"

_animals, 2020, doi:10.3390/ani11010052_

Round 1

Reviewer 1 Report

Review of “Growth, carcass and meat characteristics of grass-fed lambs weaned from extensive tussock areas and grazed on permanent pastures or alfalfa

A very topical paper that will create a lot of interest in relation to pasture reared healthy lean meat which is of interest to many in the world.

The manuscript is well written, and I have only minor suggestions for improvement

  1. Title I am not sure that tussock area should be used as you describe the botanical composition of the grassland in table 2 and in line 84 use a better word ‘ rangeland’ which is also used in other places in the manuscript
  2. Similar to above in line 32 the term extensive high-country area better term is extensive rangeland
  3. Line 87 add improved to permanent pasture
  4. Line 39 delete ‘a’ before tender meat
  5. Line 50 change to rangeland to replace tussock and throughout the manuscript
  6. Line 92 I would prefer table 1 to integrated into table 4 like tables 6 – 8
  7. Line 108 what was used to control parasites ie product and maker?
  8. Line 145 add with ‘pre slaughter and ‘live’ weights
  9. Line 170 mentions analyze texture no texture results given at all
  10. Lines 305 to 310 mention made of low crude protein levels what is required minimum protein to achieve LWG need to give a reference here i.e. above 14 % CP or something like that
  11. Line 320 to 321 The reference is for standards for Uruguay not worldwide
  12. Line 323 “high degree of finishing” what does that mean – ‘a higher fat level? Or what need to be more specific
  13. line 342 to 343 here the authors introduce a new term ie natural unimproved rangeland just be consistent ie rangeland
  14. Likewise, line 345 ‘natural pastures’ you mean rangeland
  15. Line 353 start sentence with In not On
  16. Line 357 to 359 reword these two sentences into better English ie ‘Overall high sensory attributes were detected by taste panel at weaning in comparison to those lambs grazed post weaning on permanent pasture’ and perhaps mention age of lambs at slaughter here also
  17. Line 388 delete “the assessment of the” and quality so reads “important for the marketing of grass-fed lamb meat’.
  18. Line 389 from Low to line 393 DHA I would delete it repeats results
  19. Line 396 add ‘the’ before commercial cuts
  20. Line 397 and 398 I don’t understand what the authors mean here suggest it is something like “although lamb meat at weaning (age of lamb) is high quality there is a sacrifice in term of carcass weight and hence maybe insufficient supply for potential markets “

Reviewer 2 Report

Presented for review work moves the current problem of management in sheep breeding based on extensive grazing pastures which can allowed get better quality of meat for human health.There is a lot of research work in this area but in many regions of the world a natural management based on extensive pastures plays important role in these areas, as a component of landscape, culture and supplier of many valuable products. Experimental design, lab methodology and statistical analysis are correct, discussion is interesting.My comments: Authors say that the botanical composition and chemical characteristics of pasture are  shown in Table 2 but there are only data on alfalfa and permanent pasture without extensive rangeland (before weaning).Why samples of 10 and 35 g were used for the fatty acid analyses of meat?The authors refer to the Australian and New Zealand standards to claim meat as a source of omega-3. Does it result from the systems used to a largely used in lamb production extensive finishing systems?

The article was written very carfully and  can be publish in Animals after explaining inaccuracies, which I present above.

Reviewer 3 Report

Meat quality from different production systems is always a topic of interest as it  directly impact farmers economy/revenue and consumers demands

In general the work seems to be adequately conducted but there are some issues which would need to be clarified (see attached file).

The main concern for this reviwer is related to statistical analysis. As described in methodology it seems that some variables are discrete and not continuous. However, there is not description of how data was handled to cope with this.

Further details in file

Round 2

Reviewer 3 Report

The authors have adequately answered all comment from round 1

I do not have further comments